# Unmasking LAION-5B: Age, Gender, Race, and Emotion Biases in Large-Scale Image Datasets

## Abstract

Large-scale image-text datasets, such as LAION-5B, are foundational to modern AI systems, yet their vast scale and uncurated nature raise significant concerns about demographic and stereotypical biases. This study presents a comprehensive analysis of the demographic composition and representational, stereotypical, and intersectional biases in LAION-2B-en and LAION-2B-multi, the two main components of the LAION-5B dataset. Using state-of-the-art models—FairFace, DeepFace, and Emo-AffectNet—we analyze faces detected in the dataset to identify biases across age, gender, race, and expressed emotion. Our findings reveal substantial overrepresentation of young adults (20–39), White individuals, and males, alongside consistent underrepresentation of minority racial groups and middle-aged or older women across both dataset components. We also observe stereotypical associations between demographic attributes and emotions, such as "Anger" being predominantly linked to males and "Happiness" to females, pointing to systemic imbalances in the data. The consistency of these patterns across two demographic models and both components of LAION-5B demonstrates that these biases are deeply embedded in one of the most widely-used training datasets. Given the scale at which LAION-5B is used to train generative models, these demographic imbalances could shape the behavior and outputs of numerous downstream AI systems.

## 1 Introduction

The accelerated widespread of Artificial Intelligence (AI) in recent years has brought critical ethical considerations to the forefront, with fairness and the mitigation of discrimination as key concerns (Dwivedi et al., 2021). While biases in classical AI systems and Large Language Models (LLMs) have been extensively documented, leading to the discovery of discriminatory models in sensitive domains (Berk et al., 2018; Motoki et al., 2023; Bender et al., 2021), the newer frontier of generative AI, particularly text-to-image models, presents its own profound set of challenges regarding bias (Bommasani et al., 2022; Wan et al., 2024).

The LAION-5B dataset (Schuhmann et al., 2022), with its billions of image-text pairs, has become essential for training state-of-the-art generative models. The sheer scale of such datasets, often compiled through web-scraping, influences both the capabilities and biases of the models trained upon them (Birhane & Prabhu, 2021; Birhane et al., 2021). Prior research has primarily focused on LAION-5B's harmful content or its role in producing stereotypical outputs in trained models (Birhane et al., 2023; Luccioni et al., 2023), leading to the dataset's temporary withdrawal and 2024 rerelease (LAION e.V., 2024). However, the underlying demographic composition and biases of the dataset itself remain less thoroughly explored, despite being a key bias source (Suresh & Guttag, 2021; Ntoutsi et al., 2020).

There are significant fairness implications of using vast, often uncurated, datasets like LAION-5B for training generative AI. These datasets act as a primary source for bias, codifying societal prejudices which are then learned and potentially amplified by the models (Suresh & Guttag, 2021; Ntoutsi et al., 2020; Nicoletti & Bass, 2023). Imbalances in demographic group prevalence (*representational bias*), unwarranted associations between demographic attributes and other characteristics (*stereotypical bias*, e.g., between gender and emotion) or unwarranted associations between multiple demographic attributes (*intersectional bias*, e.g., between gender and age) in the training data can

cause generative models to produce visual content that reinforces harmful societal biases (Nicoletti & Bass, 2023; Abbasi et al., 2019; Lloyd, 2018; Dominguez-Catena et al., 2024c).

This paper addresses the challenge of analyzing demographic bias within the LAION-5B dataset. Given LAION-5B's size and the absence of explicit demographic information, an exhaustive manual annotation is infeasible. Therefore, our methodology analyzes LAION-5B (LAION e.V., 2024) through automatic inference of demographic attributes from faces detected in the dataset. We employ multiple models to detect faces and estimate age, race, gender, and emotion, using Fair-Face (Karkkainen & Joo, 2021) and DeepFace (Serengil & Ozpinar, 2024; 2020) for demographic attributes and Emo-AffectNet (Ryumina et al., 2022) for facial expressions. The inclusion of emotions alongside the more traditional demographic attributes is motivated by prior studies that have identified biases in internet-sourced datasets used for Facial Expression Recognition (Dominguez-Catena et al., 2024a). This analysis is carried out separately for the two main LAION-5B components to analyze whether the multilingual nature of LAION-2B-multi provides greater demographic diversity compared to the English-only LAION-2B-en.

By focusing on the foundational demographic composition of LAION-5B, this work aims to provide insights into the biases that generative models may inherit, complementing existing research that focuses on downstream model outputs or harmful content (Birhane et al., 2021; LAION e.V., 2024; Birhane et al., 2023; 2024; Luccioni et al., 2023).

Our analysis revealed several key demographic and emotional biases in the LAION-5B dataset, summarized as follows:

- Significant overrepresentation of young adults (20–39), White individuals, and males in both LAION-2B-en and LAION-2B-multi datasets.
- Strong stereotypical biases in facial expressions, with emotions like "Anger" and "Disgust" disproportionately associated with males and "Happiness" with females.
- Alignment of these biases in both LAION-2B-en and LAION-2B-multi and across two demographic models, FairFace and DeepFace, reinforcing their systemic nature.
- High similarity between the demographic compositions of LAION-2B-en and LAION-2B-multi datasets. Some minor differences in LAION-2B-multi are the greater representation of older individuals, increased disparity in gender representation, and greater racial diversity.

In summary, the main contributions of this work are:

- A detailed demographic analysis of a substantial sample (1,000,000 image URLs) of the LAION-2B-en and LAION-2B-multi datasets, focusing on age, gender, race, and emotion using FairFace, DeepFace and Emo-AffectNet.
- An intersectional and stereotypical bias analysis leveraging Ducher's Z metric to uncover co-occurrence patterns across demographic attributes and associations between demographic and emotion categories.

## 2 BACKGROUND

### 2.1 FAIRNESS AND DISCRIMINATION IN GENERATIVE AI

The increasing integration of Artificial Intelligence (AI) into societal structures has raised significant ethical concerns, particularly regarding fairness and bias mitigation (Dwivedi et al., 2021; Christoforaki & Beyan, 2022). Bias in "classical" AI systems for classification and risk assessment has been extensively documented (Mehrabi et al., 2021; Pessach & Shmueli, 2020), demonstrating discriminatory outcomes in critical domains including criminal justice (Berk et al., 2018; Avella, 2020) and employment hiring (Dastin, 2018). Large Language Models (LLMs) have similarly been shown to harbor and propagate significant biases, including political leanings (Motoki et al., 2023; Buyl et al., 2024), gender stereotypes (Garrido-Muñoz et al., 2023), and cultural misrepresentations (Naous et al., 2024). Such biases often originate from the vast, frequently uncurated datasets on which these models are trained (Bender et al., 2021; Chang et al., 2023).

Generative AI, particularly text-to-image models, represents a newer frontier where bias implications are profound and actively investigated (Bommasani et al., 2022; Wan et al., 2024), as these models can perpetuate and amplify societal stereotypes by generating images that reinforce traditional gender roles or racial caricatures (Nicoletti & Bass, 2023; Luccioni et al., 2023; Cheong et al., 2024). The rapid development of influential models like DALL-E (Ramesh et al., 2021) and Stable Diffusion (Rombach et al., 2022) underscores the critical need to address these embedded biases.

## 2.2 DATASET BIAS

Training data is widely recognized as a primary source of AI bias, acting as a bottleneck where biases from collection, sampling, and annotation converge and are codified (Suresh & Guttag, 2021; Ntoutsi et al., 2020). These biases often reflect historical and societal prejudices rather than mere statistical anomalies (Fabris et al., 2022; Dominguez-Catena et al., 2024b). Understanding and quantifying dataset bias is therefore critical for building fairer AI systems (Dominguez-Catena et al., 2024a).

Three common types of dataset bias are representational, stereotypical, and intersectional bias (Dominguez-Catena et al., 2024a; Buolamwini & Gebru, 2018; Kang et al., 2021). **Representational bias** refers to unbalanced representation of demographic groups, degrading model performance for under-represented populations (Mehrabi et al., 2021). **Stereotypical bias** arises when specific attributes are systematically associated with certain demographic groups, reflecting and reinforcing societal stereotypes (Abbasi et al., 2019; Bordalo et al., 2016; Dominguez-Catena et al., 2024c). **Intersectional bias** emerges at the intersection of multiple demographic attributes in ways that differ from biases affecting each attribute independently (Buolamwini & Gebru, 2018; Kang et al., 2021). Training models on such data leads to learning these correlations, resulting in unfair predictions and amplifying existing societal inequalities (Lloyd, 2018), making data curation quality crucial (Shome et al., 2022).

## 2.3 BIAS IN LARGE IMAGE DATASETS

Large-scale image datasets foundational to modern computer vision and generative AI commonly inherit and concentrate online biases due to their web-scraping origins (Birhane & Prabhu, 2021; Fabbrizzi et al., 2022). The popular ImageNet (Deng et al., 2009) dataset shows problematic categorizations in its "person" subtree and demographic imbalances (Yang et al., 2020; Denton et al., 2021; Dulhanty & Wong, 2019), while MSCOCO (Lin et al., 2015) shows biases in depicting people and activities that propagate to downstream tasks such as image captioning (Zhao et al., 2021). These analyses typically focus on label quality, harmful content, or performance disparities of trained models.

The popular LAION-5B dataset (Schuhmann et al., 2022) has been scrutinized for biases, with previous research highlighting harmful content, misogyny, and malignant stereotypes (Birhane et al., 2021; LAION e.V., 2024), as well as the tendency of models trained on it to amplify societal biases and generate stereotypical imagery (Birhane et al., 2023; 2024; Luccioni et al., 2023). Some of these findings led to public removal of the dataset in 2023, followed by release of a clean version in 2024 (LAION e.V., 2024), which we use in this work. While these studies provide critical insights into LAION-5B's content and downstream effects, this paper focuses specifically on the demographic composition and balance within the dataset, building on previous methodology (Dominguez-Catena et al., 2024a) for facial expression recognition (FER) datasets. This focus on inherent demographic balance is distinct yet complementary to prior analyses of harmful content or model output bias. Understanding demographic imbalances at the dataset level is crucial, as they can fundamentally skew the "worldview" learned by generative models, directly impacting fairness and representativeness of generated content (Nicoletti & Bass, 2023; Wan et al., 2024).

## 3 METHODOLOGY

### 3.1 SOURCE DATASET

Our study utilizes data from the LAION-5B project, specifically the LAION-2B-en (English) and LAION-2B-multi (multilingual) components[1] (LAION e.V., 2024). We initially sampled 501,147 URLs from LAION-2B-en and 503,130 from LAION-2B-multi, for a total of 1,004,277 URLs.

From the original ∼1 million URL sample we successfully retrieved 227,748 images from the English partition and 236,413 from the multilingual partition, for a total of 464,161 images. The content of each image was hashed and verified against the information in the LAION-5B dataset to ensure data integrity. We then employed the RetinaFace face detection system (Deng et al., 2020; Serengil & Ozpinar, 2020) to identify human faces. To ensure sufficient quality for demographic analysis, faces with a resolution below $48 \times 48$ pixels were discarded. This filtering process yielded a final sample of 37,331 faces from the English partition and 42,571 from the multilingual partition, totaling 79,902 faces for our analysis.

While our sample of ∼1 million URLs represents a small fraction of the full dataset (∼0.02%), it was chosen to be sufficiently large for a representative analysis while remaining computationally feasible. To establish the statistical reliability of our findings, we calculated the margin of error (MOE) for the final image sample. Considering membership in any demographic group as a binary attribute, the worst-case MOE for any reported proportion can be estimated. For our smallest subsample ($n = 37,331$ faces from LAION-2B-en), the formula $MOE_\gamma = z_\gamma \times \sqrt{p(1-p)/n}$ yields a margin of error of $\pm 0.51\%$ at a $95\%$ confidence level ($\gamma = 0.95$) for the worst-case scenario ($p = 0.5$). This level of precision supports the validity of the demographic proportions reported in our results.

Table 1: Dataset subsample sizes

|  | Attempted | Downloaded | Faces |
|---|---|---|---|
| relaion2B-en-research | 501,147 | 227,748 | 37,331 |
| relaion2B-multi-research | 503,130 | 236,413 | 42,571 |
| Total | 1,004,277 | 464,161 | 79,902 |

### 3.2 DEMOGRAPHY AND EMOTION RECOGNITION

For each detected face, we extract demographic attributes using two complementary models to ensure robustness: FairFace (Karkkainen & Joo, 2021), designed for robust multi-demographic classification, and DeepFace (Serengil & Ozpinar, 2024; 2020), a widely used framework for facial attribute analysis. Both models predict age, race, and gender. To enable consistent comparison, we align their output categories by merging FairFace's Southeast Asian and East Asian categories into a single "Asian" category matching DeepFace's classification, and segmenting DeepFace's integer age predictions into FairFace's age ranges (0–2, 3–9, 10–19, 20–29, 30–39, 40–49, 50–59, 60–69, and 70+). We also analyze facial expressions using Emo-AffectNet (Ryumina et al., 2022), a model trained on a large combination of datasets.

While these tools are not perfect, they are generally robust: DeepFace reports an age MAE of $\pm 4.65$ years and gender accuracy of 97.44% on IMDB-WIKI (Rothe et al., 2015), and 68% race accuracy on FairFace (Karkkainen & Joo, 2021); FairFace reports stronger performance on its benchmark, with race accuracy of 93.7% for White and 75.4% for non-White groups, gender accuracy of 94%, and age-group accuracy of 60% (Karkkainen & Joo, 2021). For facial expressions, Emo-AffectNet obtains 66.4% accuracy on the AffectNet test set (Mollahosseini et al., 2019).

---

[1]In this paper we use the 2024 rerelease, in particular the *relaion2B-en-research* (`https://huggingface.co/datasets/laion/relaion2B-en-research`) and *relaion2B-multi-research* (`https://huggingface.co/datasets/laion/relaion2B-multi-research`) partitions.

### 3.3 Bias analysis

Using demographic predictions from FairFace and DeepFace, alongside expression data from Emo-AffectNet, we analyze proportions of each demographic group and expression category to assess **representational bias**. We also conduct **intersectional bias** analysis of demographic attributes using Ducher's Z metric (Ducher et al., 1994), following recommendations in (Dominguez-Catena et al., 2024a), and **stereotypical bias** analysis between demographic attributes and recognized emotion using the same metric. Z compares observed co-occurrence of group $g \in G$ and class $y \in Y$ to expected co-occurrence if independent, defined as:

$$\text{Z}(X, g, y) = \begin{cases} \frac{p_{g \wedge y} - p_g p_y}{\min[p_g, p_y] - p_g p_y} & \text{if } p_{g \wedge y} - p_g p_y > 0 \\ \frac{p_{g \wedge y} - p_g p_y}{p_g p_y - \max[0, p_g + p_y - 1]} & \text{if } p_{g \wedge y} - p_g p_y < 0 \\ 0 & \text{otherwise,} \end{cases} \tag{1}$$

where $p_g$, $p_y$ and $p_{g \wedge y}$ are the proportions of samples in population $X$ belonging to group $g$, class $y$ or both, respectively. For intersectional bias, class $y$ can be replaced by a second demographic group $g' \in G'$. The values of Z range from $-1$ (maximum underrepresentation) to $1$ (maximum overrepresentation), with $0$ indicating no correlation.

## 4 Results

### 4.1 Representational bias. Demographic distribution

Fig. 1 shows the demographic composition of LAION-2B-en (blue) and LAION-2B-multi (orange) according to both FairFace and DeepFace.

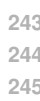
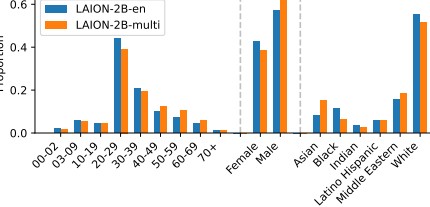
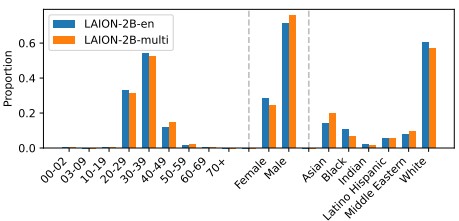

(a) Demographic profile according to FairFace. (b) Demographic profile according to DeepFace.

Figure 1: Distribution of demographic attributes (age, gender, race) in LAION-5B.

**Age.** Both models indicate strong overrepresentation of 20–39-year-olds. FairFace places more data in 20–29, while DeepFace shifts toward 30–39. According to both models, LAION-2B-en skews slightly younger than LAION-2B-multi, with higher representation of individuals under 40.

**Gender.** Both models agree on male predominance in the dataset. FairFace estimates 57 to 61% male, while DeepFace exceeds 70%. Despite scale differences, the direction is consistent, and both models agree on a higher discrepancy in favor of males in LAION-2B-multi.

**Race.** Both models identify White as the largest group (50 to 60%), far above a balanced 16% baseline. Differences appear in minority categories: FairFace reports more Middle Eastern and fewer Asian individuals; DeepFace reports the opposite. Both agree in the underrepresentation of Black, Indian, and Latino Hispanic groups.

**Overall.** While there are some minor differences between demographic estimations from FairFace and DeepFace, LAION-2B-en and LAION-2B-multi exhibit similar profiles: concentration in ages 20–39, male overrepresentation, and a White majority. These patterns indicate substantial representational imbalances across all components.

## 4.2 REPRESENTATIONAL BIAS. EMOTION DISTRIBUTION

Fig. 2 shows the facial expression distribution identified by Emo-AffectNet, dominated by "Happiness" and "Neutral," with "Fear," "Disgust," and "Surprise" comparatively rare. This mirrors common internet-sourced FER datasets, such as AffectNet (Mollahosseini et al., 2019; Dominguez-Catena et al., 2024a), suggesting that LAION-5B follows broader online trends. There are minor differences between LAION-2B-en and LAION-2B-multi, with the English variant including more "Happy" examples, while the multilingual variant favors "Neutral" and "Sadness".

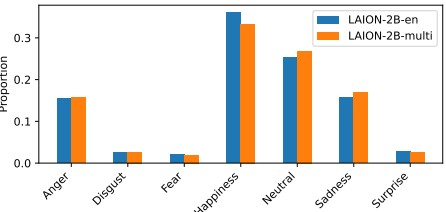

Figure 2: Distribution of facial expressions in LAION-5B, according to Emo-AffectNet.

## 4.3 INTERSECTIONAL BIAS

We quantify intersectional bias using Duchers Z (Ducher et al., 1994) across age, gender, and race (Fig. 3). Groups below 1% prevalence are excluded, as Z scores for these edge cases become unreliable. As Sections 4.1 and 4.2 showed near-identical component profiles, we report results on the aggregated dataset.

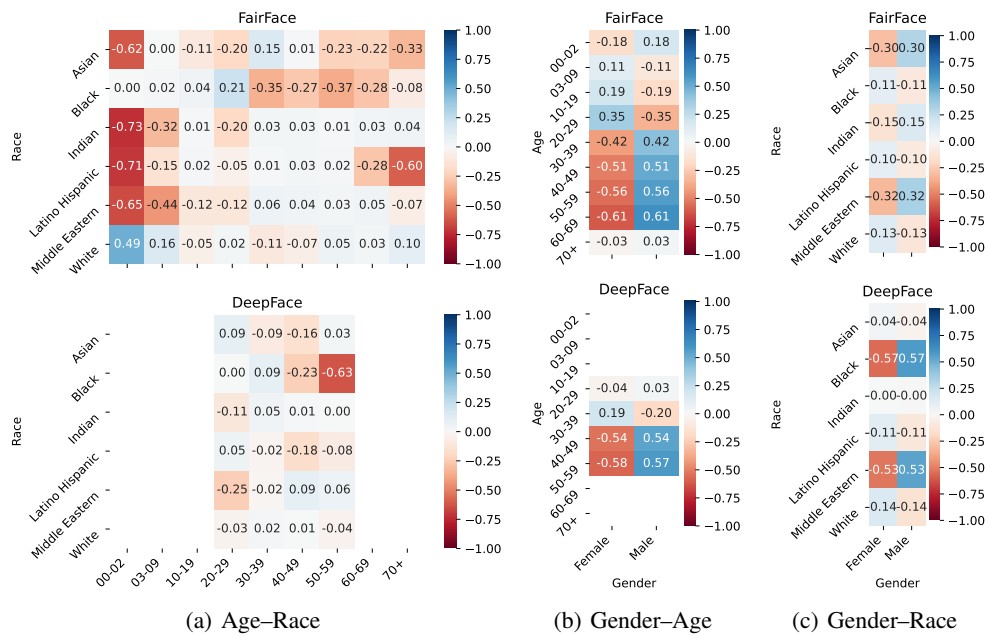

(a) Age–Race  (b) Gender–Age  (c) Gender–Race

Figure 3: Intersectional bias (Duchers Z) across demographic attribute pairs.

**Age–Race.** FairFace indicates underrepresentation of the oldest groups (60+) across most races, and of very young Asian, Indian, Latino Hispanic, and Middle Eastern children; while White infants are relatively overrepresented. DeepFace shows sparser age coverage and weaker biases, but echoes the underrepresentation of older Black individuals, aligning with FairFace on this pattern.

**Gender–Age.** The gender-age analysis reveals strong and consistent biases across both FairFace and DeepFace. FairFace shows female overrepresentation below 30 and male overrepresentation above 30. DeepFace exhibits the same pattern with the threshold shifted to the age of 40, consistent with its older age estimates overall.

**Gender–Race.** Biases are less consistent in the gender–race analysis, with only the underrepresentation of Middle Eastern females shared by both models. DeepFace also indicates the underrepresentation of Black females, but FairFace instead indicates a slight overrepresentation of this group.

**Overall.** Gender–age pairings exhibit the strongest consistent biases, with younger females and older males overrepresented across models; age–race and gender–race biases are present but generally smaller and with less consistency between models.

## 4.4 FACIAL EXPRESSION STEREOTYPICAL BIAS

We examine stereotypical biase between emotion and demographic attributes via Duchers Z (Fig. 4). Groups with a representation of less than 1% of the dataset are excluded to ensure the stability of Z score measures. Results are given for the aggregated dataset, composed of LAION-2B-en and LAION-2B-multi.

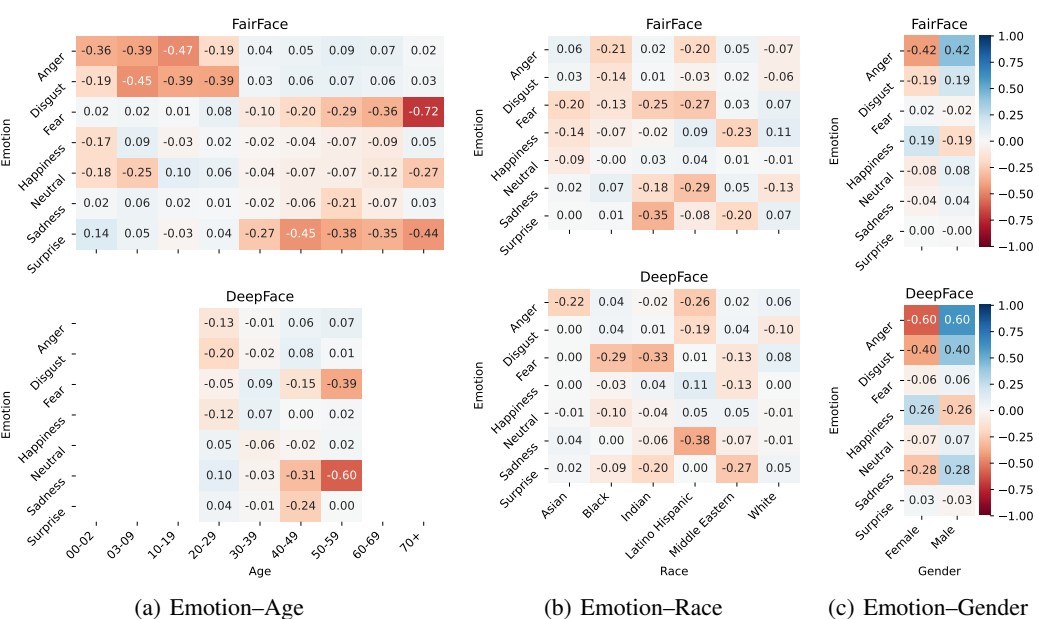

(a) Emotion–Age      (b) Emotion–Race      (c) Emotion–Gender

Figure 4: Emotion bias (Duchers Z) across demographic pairs.

**Emotion–Age.** Stereotypical biases between emotion and age show some consistent patterns, despite being weak overall. The strongest biases are the underrepresentation of age groups under 30 years in "Anger" and "Disgust"; and the underrepresentation of older groups in "Fear", "Sadness" and "Surprise".

**Emotion–Race.** Effects on emotion–race are subtle and model-dependent. Recurring patterns include lower representation in "Fear" and "Surprise" for Indian individuals, lower representation in "Anger" and "Sadness" for Latino Hispanic individuals and lower representation in "Surprise" for Middle Eastern individuals.

**Emotion–Gender.** Emotion–gender shows stronger stereotypical biases compared to the other two demographic attributes. Both models indicate overrepresentation of males in "Anger" and "Disgust," while females are overrepresented in "Happiness."

**Overall Conclusions.** Emotion–demographic associations are weaker but reveal consistent stereotypical biases: males are linked to "Anger" while females are linked to "Happiness", younger individuals are linked to negative emotions (e.g., "Anger," "Disgust") and certain racial groups are disproportionately associated with certain emotions (e.g., Indian with "Fear" and "Surprise"). These patterns persist across FairFace and DeepFace, indicating biases strongly embedded in the dataset.

## 5 STATISTICAL ROBUSTNESS OF BIAS ANALYSIS

To validate the statistical robustness of our associational bias analysis, we focus here on the Ducher's Z findings for intersectional and stereotypical bias. The stability of the representational bias analysis is already supported by the large sample size and its corresponding low margin of error, as established in Section 3.1.

To validate the Ducher's Z scores, we employed a bootstrap resampling technique over $1,000$ repetitions to construct $95\%$ confidence intervals (CIs). An observed association is considered statistically significant if its 95% CI is narrow and does not cross zero. For brevity, CIs in this section are reported for FairFace only, and only for the key results.

Our analysis confirms the stability of our most salient conclusions. For instance, the "angry-man-happy-woman" stereotype is strongly supported: the Z-score for the over-representation of males with "Anger" $(0.42)$ yielded a 95% CI of $[0.4, 0.43]$, while the female-"Happiness" association $(0.19)$ had a CI of $[0.18, 0.2]$. Intersectional biases also proved highly stable. The over-representation of females in the 20–29 age group $(0.35)$ was confirmed with a CI of $[0.34, 0.35]$, and the significant under-representation of Middle Eastern females $(-0.32)$ was validated with a CI of $[-0.34, -0.3]$. The narrowness and consistent sign of these intervals provide strong evidence that the reported biases are significant features of the dataset.

## 6 DISCUSSION

Our analysis reveals significant demographic and stereotypical biases in LAION-2B-en and LAION-2B-multi, mirroring patterns found in other large-scale, Internet-sourced datasets, such as those intended for FER (Dominguez-Catena et al., 2024a). Both components exhibit strong representational biases toward young adults (20–39 age range), White individuals (50–60%), and males (57–70%), with some variations between FairFace and DeepFace predictions. These biases align with those previously identified in image generation models trained on LAION-5B (Nicoletti & Bass, 2023), while the pronounced male overrepresentation contrasts with the generally balanced gender distribution observed in FER datasets (Dominguez-Catena et al., 2024a). This imbalance raises concerns regarding potential reinforcement of gender disparities in downstream applications.

The intersectional analysis highlights critical disparities, with particularly strong and consistent biases observed in the gender-age pairing. Younger females (under 30 in FairFace and under 40 in DeepFace) and older males are consistently overrepresented, while middle-aged and older women are significantly underrepresented. These gender-age biases are compounded by age-race disparities, where younger and older individuals across all minority racial groups—all except White—are often underrepresented. White infants, in particular, are disproportionately overrepresented compared to infants from other racial groups. These patterns reflect entrenched societal stereotypes and imbalances that could adversely affect fairness in AI applications (Dominguez-Catena et al., 2024b).

The analysis of stereotypical biases between facial expressions and demographic attributes reveals weaker but consistent patterns, especially regarding gender. Males are more strongly associated with "Anger" and "Disgust," while females are linked to "Happiness," echoing familiar gendered stereotypes such as "angry-man-happy-woman" (Becker et al., 2007). Racial biases are subtler but suggest disproportionate underrepresentation of certain emotions, such as "Fear" and "Sadness," among Latino Hispanic and Indian individuals. These findings emphasize the need for caution when using LAION-5B datasets in both FER tasks and general image generation, as they risk perpetuating harmful stereotypes.

The consequences of these dataset biases on trained models are highly context-dependent and can vary markedly across applications. Prior work shows that for some tasks—such as facial-expression recognition—demographic imbalances have only modest effects, whereas stereotypical biases can

strongly influence model predictions (Dominguez-Catena et al., 2025). In generative AI systems, however, such biases may have far greater impact (Nicoletti & Bass, 2023). Given the widespread usage of LAION-5B, if these biases propagate to downstream models they could further amplify societal biases in the content they produce.

## 6.1 LIMITATIONS

Our analysis has several limitations. First, it relies on auxiliary models—FairFace, DeepFace, and Emo-AffectNet—for face detection and demographic and emotion classification, which can introduce their own biases into our measurements. To improve reliability, we use two independent models (FairFace and DeepFace) for demographic attributes and compare their outputs; however, these models may share systematic biases that could leak into our estimates. Disentangling dataset-intrinsic bias from tool-induced bias will require validation against self-reported demographics or human-labeled annotations in future works.

Second, our demographic categories are constrained by the models' predefined labels. These labels ignore some aspects of human diversity, especially nuanced gender identities and complex racial and ethnic categories beyond conventional taxonomies.

Third, we intrinsically treat balanced composition as desirable, yet fairness is context-dependent and admits multiple definitions (Mitchell et al., 2021). No single dataset mix is universally optimal, and perfect demographic parity may not always yield the fairest models.

Finally, dataset imbalances do not map straightforwardly to model bias (Dominguez-Catena et al., 2025). Demographic representation offers only a proxy and support tool for understanding downstream fairness, not a precise prediction.

## 7 CONCLUSIONS

This study provides a comprehensive analysis of the demographic and stereotypical biases present in the LAION-2B-en and LAION-2B-multi datasets. Our findings reveal biases such as strong overrepresentation of young adults, White individuals, and males, alongside consistent underrepresentation of minority racial groups and middle-aged or older women. These biases appear in both dataset components, with the multilanguage component being only slightly more diverse regarding race and age, at the cost of decreased gender diversity. Furthermore, stereotypical biases in facial expressions were observed when analyzing the full dataset, with males frequently associated with negative emotions such as "Anger" and "Disgust" and females with positive emotions like "Happiness." While most of these biases mirror patterns seen in other datasets (Dominguez-Catena et al., 2024a), LAION-5B's pronounced gender bias raises unique concerns.

These results uncover pervasive biases in LAION-5B's composition, affecting multiple demographic and non-demographic attributes through both general group representation and inter-group associations. Addressing these issues requires careful training dataset curation and development of more inclusive demographic analysis tools. Future work should validate these findings with human-labeled data and investigate downstream impacts on image generators such as Stable Diffusion and FLUX, as well as emerging multimodal models. Additionally, future studies could replicate this analysis on alternative datasets like COYO-700M or RedCaps and enhance robustness by incorporating diverse demographic-prediction models. Other characteristics could be studied, such as socioeconomic status, androgyny, body size, religious attire presence or capture country. Finally, given video production's substantially higher costs than image production, video datasets and generation models may exhibit even stronger demographic biases.

## 8 ACKNOWLEDGEMENTS

The authors acknowledge the use of Claude Sonnet 4 and Gemini 2.5 Pro models for language review and polishing. All generated content was directly reviewed, edited, and verified by the authors, who take full responsibility for the final published work.

Funding information anonymized for review.

## REPRODUCIBILITY STATEMENT

All code, derived data, and scripts required to reproduce the analyses, figures, and tables in this paper are provided in the supplementary material as a compressed ZIP archive.

The original image data is sourced from the publicly available LAION-5B dataset. Due to the scale of the dataset and licensing restrictions, we do not redistribute the images. However, to facilitate full replication of our findings, we provide comprehensive CSV files containing the 79,902 image URLs from our sample, along with the complete set of estimated demographic attributes (gender, age, race) and perceived emotions for each face detected.

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
