# OpenReview forum: "Unmasking LAION-5B: Age, Gender, Race, and Emotion Biases in Large-Scale Image Datasets"
_ICLR.cc/2026/Conference — Submitted to ICLR 2026_

### Official Review · Reviewer_njoF · 2025-10-22

**Soundness:** 2
**Presentation:** 2
**Contribution:** 2
**Rating:** 2
**Confidence:** 3

**Summary:**

- The paper addresses a critical point in the field of generative AI: large-scale uncurated image-text datasets (e.g., LAION-5B) serve as foundational training data for modern AI systems, but their inherent biases remain insufficiently explored, potentially distorting downstream model fairness.
- The paper systematically analyze LAION-5B’s two main components (LAION-2B-en and LAION-2B-multi), and then has the several conclusions : 1) Both dataset components exhibit significant representational biases, with overrepresentation of 20–39-year-olds, White individuals, and males, and underrepresentation of minority racial groups and middle-aged/older women; 2) Strong stereotypical biases exist (e.g., males linked to “Anger”/“Disgust,” females to “Happiness”); 3) Gender-age intersectional biases are the most consistent (younger females and older males overrepresented); 4) LAION-2B-multi shows slight improvements in race/age diversity but worse gender balance.
- The paper conclude that these biases are deeply embedded in LAION-5B and may propagate to downstream generative AI systems.

**Strengths:**

1. The paper systematically examines three key types of biases (representational, intersectional, stereotypical) using well-validated tools and metrics. This design ensures the reliability of bias detection, as evidenced by consistent results across both models and dataset components.
2. Prior studies on LAION-5B focused on harmful content or model output bias, while this paper targets the under-explored “dataset intrinsic demographic composition” . It explicitly links dataset biases (e.g., male overrepresentation, gender-emotion stereotypes) to potential risks in downstream generative models (e.g., may reinforce gender disparities), complementing existing research and providing a foundational reference for dataset curation in fair AI.

**Weaknesses:**

* 1. The paper acknowledges reliance on FairFace, DeepFace, and Emo-AffectNet but lacks in-depth analysis of how these models’ own biases may confound results.
* 2. The paper states that “balanced composition is treated as desirable” (Section 6.1) but does not explore alternative fairness definitions. For example, it does not explain whether a “balanced gender ratio” is necessary for LAION-5B, given that some generative tasks (e.g., medical image generation) may require domain-specific demographic distributions. This leads to a one-sided fairness framing. An improvement direction is to add a subsection in Discussion: “Contextual Fairness Considerations for LAION-5B,” contrasting demographic balance with task-specific fairness goals.

**Questions:**

>1. The paper uses ~1 million URLs (0.02% of LAION-5B) and justifies it via MOE (Section 3.1), but LAION-5B is web-scraped, which may have geographic/language-based data clustering (e.g., LAION-2B-multi has non-English content). Is there evidence that the sampled URLs are distributed uniformly across geographic regions or language families? For example, does the LAION-2B-multi sample include sufficient content from low-resource languages to represent their demographic characteristics? This is critical to validating whether sample biases reflect the full dataset.
>2. The paper excludes groups with <1% prevalence (in Section 4.3) due to unreliable Z scores. May this exclude marginalized subgroups (e.g., elderly Black females) that are critical for fair AI. Does the paper have plans to expand the sample size or use alternative statistical methods to analyze these low-prevalence groups?
>3. **The paper suggests “careful training dataset curation” (Section 7) but provides no specific actionable strategies. How to address the identified biases? Providing such guidance would enhance the paper’s practical value.**

---

> ### Author Response · Authors · 2025-12-03
>
> We thank the reviewer for recognizing the systematic nature of our analysis and for noting that our focus on dataset-intrinsic demographic composition complements prior work on harmful content and model outputs.
>
> Regarding tool-induced biases, we acknowledge this limitation explicitly in Section 6.1. The use of two independent demographic models was designed to mitigate this concern, and the consistency of our main findings across both supports their validity.
>
> On the fairness framing, we appreciate this nuanced point. We do acknowledge in Section 6.1 that "fairness is context-dependent and admits multiple definitions" and that "no single dataset mix is universally optimal." The reviewer's suggestion to expand this discussion with task-specific considerations is valuable and something we will consider for future versions.
>
> Concerning sample representativeness across geographic and language distributions, this is a fair question. Our random sampling from the official dataset partitions should yield a representative subset, though we acknowledge that validating geographic/language coverage explicitly would strengthen the analysis.
>
> On excluding low-prevalence groups, we agree this is a limitation. The 1% threshold was necessary for statistical stability of Z scores, but we recognize this may obscure biases affecting the most marginalized subgroups. Larger sample sizes or alternative methods could address this in future work, but would add noise to the current experimentation.
>
> Finally, regarding actionable mitigation strategies, we intentionally aim our contribution to bias quantification rather than mitigation, as the latter represents a distinct research direction. However, we believe that detailed bias characterization (as we provide) is a necessary first step that enables informed curation decisions.

---

### Official Review · Reviewer_VW3t · 2025-10-27

**Soundness:** 3
**Presentation:** 3
**Contribution:** 1
**Rating:** 2
**Confidence:** 4

**Summary:**

The authors analyze demographic and emotional bias to approximately 80K human images from two LAION-5B datasets, using pretrained computer vision models from FairFace, DeepFace, and Emo-AffectNet. They estimate demographic attributes (age, gender, race) and emotion labels, and report representational, intersectional, and stereotypical bias statistics.

**Strengths:**

The paper presents a rigorous statistical analysis of multiple bias dimensions (representational, intersectional, stereotypical).

**Weaknesses:**

- The choice of models (DeepFace and Emo-AffectNet) is questionable, as they are not state-of-the-art models but rather libraries primarily intended for easy use.
- The accuracy of the models is relatively low (e.g., DeepFace reports ~68% accuracy on race and 60% on age; Emo-AffectNet achieves ~66% on emotion). These inaccuracies likely propagate into the bias estimates, making the conclusions unreliable. For instance, Figure 1 shows inconsistent gender ratios (60–40 in FairFace vs. 70–30 in DeepFace).
- The claim in the contributions section about analyzing “a substantial sample of 1,000,000 image URLs” is misleading, since the effective analyzed sample is 80K images.
- The analysis largely confirms known facts, that web-scraped datasets are biased, without providing new insights or novel mitigation strategies.

**Questions:**

1. What is the added value of this analysis, given that it reiterates the well-known fact that web-scraped datasets contain bias?
2. Are there more curated datasets currently in active use by major AI developers? To your knowledge, do companies typically rely on uncurated LAION-5B subsets for training models?
3. Given that generative models are not typically deployed in high-stakes contexts like face recognition, what are the practical implications of the observed bias?

---

> ### Author Response · Authors · 2025-12-03
>
> We thank the reviewer for recognizing the rigor of our statistical analysis.
>
> Regarding model choice, FairFace and DeepFace are widely used in the fairness literature precisely because they offer reproducible, well-documented baselines. Our goal was not to achieve state-of-the-art demographic prediction but to conduct a robust audit using established tools. The consistency of findings across two independent models supports the validity of our conclusions despite individual model limitations.
>
> On model accuracy and the divergence between FairFace and DeepFace (particularly on gender), we acknowledge this in the paper. However, the key observation is that both models agree on the direction of biases (male overrepresentation, young adult concentration, White majority), which suggests these are genuine dataset properties.
>
> Concerning the sample size claim, we sampled 1,000,000 URLs; the 80K figure reflects faces detected from the images that were still available. We believe this is clearly explained in Section 3.1, though we appreciate the feedback on potential ambiguity.
>
> On added value, while it is known that web-scraped datasets contain biases, quantifying the specific nature and magnitude of these biases in LAION-5B (including intersectional and emotion-stereotype patterns) provides concrete evidence that was previously unavailable. This enables informed decisions about dataset use and mitigation priorities.
>
> To address the reviewer's questions directly:
> 1. LAION-5B has been used to train widely-deployed models including Stable Diffusion, so understanding its biases has practical relevance.
> 2. Most models are trained behind closed doors, without disclosure of the specific dataset mixes used for their creation. Nevertheless, LAION-5B is still reported as a key part of most these mixes.
> 3. Regarding high-stakes contexts, we note that generative models are increasingly used in media, advertising, and creative industries where representational biases can cause real harm. Even if the stakes are comparatively low, the high surface of contact with society augments the risks significantly. Additionally, image generation is employed for the creation of syntetic datasets that can be used in higher risk applications (such as facial recognition).

---

### Official Review · Reviewer_Y1a3 · 2025-10-28

**Soundness:** 2
**Presentation:** 3
**Contribution:** 2
**Rating:** 2
**Confidence:** 2

**Summary:**

This study provides an analysis of the demographic and stereotypical biases present in the LAION-2B-en and LAION-2B-multi datasets. Using the FairFace, DeepFace, and Emo-AffectNet models for face detection and attribute classification, the analysis reveals biases such as a strong overrepresentation of young adults, White individuals, and males, alongside a consistent underrepresentation of minority racial groups and middle-aged or older women.

**Strengths:**

1. The fundamental idea of this paper is technically correct.
2. The paper is well written and easy to follow.
3. The rationale behind each section and the overall motivation are clearly presented and easy to understand.

**Weaknesses:**

1. The study's entire analysis is fundamentally dependent on the outputs of third-party, pre-trained models: RetinaFace for face detection, and FairFace, DeepFace, and Emo-AffectNet for demographic and emotion classification. This methodology introduces a potentially fatal flaw. These models are known to exhibit their own inherent biases. The generated biased labels compromises the accuracy and validity of the final analysis. While the authors acknowledge this limitation and attempt to mitigate it by using two independent models, this does not resolve the core issue.
2. While the auditing of large-scale datasets is a valuable service to the community, the contribution of this paper may not meet the standard of ICLR. The paper stops at quantifying these biases without proposing or testing novel methods for bias mitigation. As such, the work feels more like a descriptive technical report than a novel research contribution.
3. The paper is primarily descriptive statistical analysis, lacking deep theoretical exploration of bias origins. For the discovered bias patterns (such as the "angry male-happy female" stereotype), there is insufficient theoretical explanation of why these patterns emerge and persist in web-scraped data.
4. While the paper notes potential impacts of dataset biases on downstream models, it lacks empirical validation. The authors do not actually test how these biases propagate to the models trained on LAION-5B.
3. From Figure 1, we can see that the demographic distributions detected by FairFace and DeepFace models are quite different. What types of samples typically show disagreement between the models? These details could be analyzed in greater depth.

**Questions:**

1. The analysis is flawed because it uses biased models to measure bias, making the results unreliable.
2. The paper lacks novelty for ICLR, as it only reports on known biases without proposing any new methods.

---

> ### Author Response · Authors · 2025-12-03
>
> We thank the reviewer for acknowledging the clarity of our writing and the soundness of our approach.
>
> Regarding the use of biased models to measure bias, this is an unavoidable challenge in large-scale dataset audits. Our use of two independent models with different architectures was designed precisely to address this. The consistency of our main findings across both FairFace and DeepFace strongly suggests these are genuine dataset properties rather than classifier artifacts.
>
> On contribution and novelty, we respectfully disagree that dataset audits are mere "technical reports." Prior influential work on ImageNet and other benchmarks has followed similar descriptive methodologies and has been recognized as valuable. Rigorously documenting biases in one of the most widely-used generative AI training datasets is a meaningful contribution, even without proposing mitigation methods (which would constitute a different research direction).
>
> Concerning the divergence between FairFace and DeepFace, we discuss the main differences in Section 4.1. A more granular disagreement analysis is an interesting suggestion for future work, though the consistency of our main conclusions across models is precisely what supports their robustness.

---

### Official Review · Reviewer_4pMi · 2025-10-31

**Soundness:** 3
**Presentation:** 3
**Contribution:** 3
**Rating:** 4
**Confidence:** 4

**Summary:**

The paper audits **LAION-5B**—specifically **LAION-2B-en** and **LAION-2B-multi**—for **representational, intersectional, and stereotypical** biases along **age, gender, race, and emotion**. The authors sample ~1M URLs, successfully download ~464k images, detect **79,902** faces (≥48×48) via RetinaFace, and infer demographics with **FairFace** & **DeepFace** plus **Emo-AffectNet** for expressions. Using **Ducher’s $Z$** to quantify association biases, they find overrepresentation of **young adults (20–39)**, **White**, and **male** faces; consistent **gender–age** skew (younger females, older males); and stereotypical **emotion–gender** links (male↔Anger/Disgust, female↔Happiness). Bootstrapped CIs (FairFace) suggest these $Z$ effects are statistically stable. Code and derived CSVs (URLs + attribute predictions) are provided.

**Strengths:**

* **Methodological triangulation:** Two independent demographic models + FER model; intersectional and stereotypical analyses via a normalized $Z$.

* **Representative sampling scale:** ~**1M** URLs attempted; ~**80k** faces after quality gating; MOE quantified.

* **Consistent patterns:** Overrepresentation (20–39, White, male), robust gender–age skew, and emotion–gender stereotypes across components.

* **Statistical robustness:** Bootstrapped CIs for hallmark $Z$ findings.

* **Reproducibility:** Code + derived CSVs (URLs + predictions) to replicate figures/tables.

**Weaknesses:**

* **Tool-induced bias not disentangled.** No human-labeled audit subset to calibrate FairFace/DeepFace/Emo-AffectNet errors within LAION; sensitivity of conclusions to classifier confusion (e.g., race or age misestimates) is not quantified.

* **Sampling/attrition analysis is thin.** ~54% download failure and face-quality filtering could induce selection bias; limited evidence that final 79,902 faces remain representative of the starting partitions beyond MOE on proportions. (MOE ignores non-response mechanisms.)

* **Partial CI coverage.** CIs focus on exemplar $Z$s (FairFace only); a fuller table—spanning both demographic models and major cells—would strengthen claims.

* **Privacy surface.** Publishing per-image URLs paired with inferred sensitive attributes may raise re-identification risk even without image redistribution. The ethics discussion acknowledges risks in general, but concrete mitigations for released CSVs are not detailed.

* **Causality caveat.** The link from dataset bias to downstream generative behavior is discussed qualitatively; no bridging experiment (e.g., controlled training subset swaps) is provided.

* **Missing Related Works.** Especially in the case of face recognition bias, several related works should be incorporated to tell the complete story of bias in data and how we arrived at our current state. ** For example, but not limited to,
   - Robinson, J. P., Livitz, G., Henon, Y., Qin, C., Fu, Y., & Timoner, S. (2020). Face recognition: too bias, or not too bias?. In Proceedings of the ieee/cvf conference on computer vision and pattern recognition workshops (pp. 0-1).

I expect this would strengthen the story while better representing the start of research in this area. Especially the work mentioned above, it is just too relevant to overlook (it questions the overall completeness of this work).

**Questions:**

1. **Non-response bias.** Can you analyze **download success vs. failure** by metadata (e.g., language/domain) and report whether face-presence/quality differs materially, to bound attrition bias?

2. **Calibration subset.** Would you include a **human-labeled audit** (say, 2–5k faces) to estimate and correct classifier biases (age bin drift, race confusion) and re-report adjusted proportions/$Z$s?

3. **Dual-model reconciliation.** Where FairFace and DeepFace diverge (e.g., Asian vs Middle Eastern shares), can you provide a **consensus estimator** and uncertainty bands across models?

4. **Emotion reliability.** Given modest FER accuracy in-the-wild, how sensitive are the emotion–demographic $Z$ patterns to plausible label noise (e.g., ±10–20% symmetric/biased flips)?

5. **Downstream linkage.** Could you run a **mini ablation**: fine-tune a diffusion backbone on stratified, re-weighted subsets to test whether reducing identified imbalances measurably changes prompt-conditioned outputs?

6. **CSV privacy.** What safeguards (hashing, rate-limited access, per-row k-anonymity checks) accompany the released URL+attribute CSVs? Would you consider releasing **aggregated** counts only?

**Details Of Ethics Concerns:**

The paper infers **sensitive attributes** (race, gender, age) and releases **per-image URLs** with those labels. Even without redistributing images, this may enable re-identification or targeted scraping and could conflict with original site TOUs. Please have an ethics reviewer assess compliance and advise on **safer release protocols** (aggregated stats, hashed URLs, access controls).

---

> ### Author Response · Authors · 2025-12-03
>
> We thank the reviewer for their thorough and constructive feedback, and for recognizing the methodological strengths of our work, including the triangulation across models, the representative sampling scale, and the reproducibility materials provided.
>
> Regarding tool-induced bias, we acknowledge this limitation in Section 6.1. The use of two independent demographic models was intended to mitigate this concern, and the consistency of our main findings across both suggests that the observed biases are unlikely to be purely classifier artifacts.
>
> On sampling and attrition, the ~54% download failure is common when working with web-scraped datasets at this scale, as URLs become stale over time. While attrition could introduce selection bias, our primary claims concern the available content in LAION-5B rather than hypothetical unreachable content.
>
> The privacy concerns are thoughtful. We note that the URLs are already public (part of LAION-5B itself), and the inferred attributes are predictions from publicly available models. Nonetheless, we will reflect on aggregated-only releases for final materials.
>
> Finally, we thank the reviewer for the pointer to Robinson et al. (2020), which we will incorporate in future versions.
>
> Regarding the specific questions:
>
> 1. Non-response bias analysis by metadata (language/domain) is an interesting suggestion we had not fully explored. This would be valuable for future studies.
>
> 2. A human-labeled calibration subset would indeed strengthen the work, even if we cannot operate at the same numbers as with the automated tools. We note this as an important direction for future validation.
>
> 3. A formal consensus estimator across models is a reasonable suggestion. Unfortunately, with only two models a per-sample consensus becomes complicated to keep statistically significant, potentially introducing excesive selection bias. For this reason, our work only deals with aggregated measures.
>
> 4. Sensitivity analysis under simulated label noise is an interesting robustness check that we did not perform, though we note that the bootstrap CIs do capture sampling variability.
>
> 5. Downstream ablation experiments (fine-tuning diffusion models on re-weighted subsets) represent a substantially different and computationally intensive research direction that we believe merits its own dedicated study, outside of the scope of our current work.
>
> 6. For CSV privacy, we indeed only intend to publish aggregated counts. Unfortunately, we note that the URLs and models are already publicly available, making this work relatively easy to reproduce.

---

### Meta-Review · Area_Chair_BCGe · 2026-01-06

**Summary:**

This paper conducts a large-scale empirical audit of the LAION-5B dataset by analyzing its two major subsets, LAION-2B-en and LAION-2B-multi, with a focus on demographic, representational, stereotypical, and intersectional biases. Leveraging existing face detection and attribute recognition models, the study examines biases across age, gender, race, and expressed emotion in images containing human faces. However, all reviewers raised substantial concerns about the validity of the methodology and the reliability of the resulting conclusions. A central issue is that the analysis fundamentally depends on third-party pretrained models, whose intrinsic biases cannot be disentangled from the dataset biases under study, making it difficult to attribute the observed effects to LAION itself. Reviewers 4pMi and njoF also raised concerns about the sampling procedure itself, noting that the collected subset may already exhibit inherent biases introduced during data downloading. In particular, they questioned whether the sampled data might be implicitly clustered by factors such as geographic region or language, which could further confound the analysis and limit the validity of the conclusions. These concerns were not sufficiently addressed in the rebuttal, and no reviewer revised their score. As a result, the overall recommendation leans toward rejection.

**Reviewer Concerns:**

Some of the issues were partially addressed in the rebuttal, but several critical concerns remain unresolved.

Most importantly, the rebuttal does not sufficiently address the central concern raised by all reviewers regarding the disentanglement of dataset bias from model-induced bias. The authors argue that using two independent models mitigates this problem; however, this assumption is not sufficiently justified, as employing multiple models does not in itself eliminate or control for the intrinsic biases present in those models.

In addition, concerns regarding potential biases introduced during data sampling remain unresolved. In the rebuttal, the authors acknowledge that the sampled data may exhibit language- and geography-based aggregation effects.

**Reviewer Scores:**

Each reviewer tends to keep their original score.

---

### Decision · Program_Chairs · 2026-01-26

Reject